# A Comprehensive Spark-Based Layer for Converting Relational Databases to NoSQL

**Manal A. Abdel-Fattah** ⬤ **, Wael Mohamed \*** ⬤ **and Sayed Abdelgaber**

Department of Information Systems, Faculty of Computers and Artificial Intelligence, Helwan University, Cairo 11795, Egypt; mafattah@fci.helwan.edu.eg (M.A.A.-F.); sgaber@fci.helwan.edu.eg (S.A.)
\* Correspondence: waelmohamed@fci.helwan.edu.eg

**Abstract:** Currently, the continuous massive growth in the size, variety, and velocity of data is defined as big data. Relational databases have a limited ability to work with big data. Consequently, not only structured query language (NoSQL) databases were utilized to handle big data because NoSQL represents data in diverse models and uses a variety of query languages, unlike traditional relational databases. Therefore, using NoSQL has become essential, and many studies have attempted to propose different layers to convert relational databases to NoSQL; however, most of them targeted only one or two models of NoSQL, and evaluated their layers on a single node, not in a distributed environment. This study proposes a Spark-based layer for mapping relational databases to NoSQL models, focusing on the document, column, and key–value databases of NoSQL models. The proposed Spark-based layer comprises of two parts. The first part is concerned with converting relational databases to document, column, and key–value databases, and encompasses two phases: a metadata analyzer of relational databases and Spark-based transformation and migration. The second part focuses on executing a structured query language (SQL) on the NoSQL. The suggested layer was applied and compared with Unity, as it has similar components and features and supports sub-queries and join operations in a single-node environment. The experimental results show that the proposed layer outperformed Unity in terms of the query execution time by a factor of three. In addition, the proposed layer was applied to multi-node clusters using different scenarios, and the results show that the integration between the Spark cluster and NoSQL databases on multi-node clusters provided better performance in reading and writing while increasing the dataset size than using a single node.

**Keywords:** big data; NoSQL; Spark; relational database; migration; transformation

## 1. Introduction

In recent decades, relational databases (RDBs) have been widely used by various enterprises. The RDB uses structured query language (SQL). It is not suitable for big data because it has three forms of data: structured, semi-structured, and unstructured, whereas RDB provides only structured and has limited capability with semi-structured and unstructured data. RDB can also store large amounts of data; however, it exhibits significant scalability, availability, and flexibility problems [1].

A new category of databases, called not only structured query language (NoSQL), has been proposed to overcome the limitations of RDB. NoSQL provides horizontal scaling, complex data storage, fault tolerance, and a high availability. Therefore, it is suitable for big data and cloud computing applications [2,3]. NoSQL databases complement the relational databases. There are four types of NoSQL databases: document, column, key–value, and graph. Spark is a large-scale data-processing engine. Spark is an in-memory processing engine designed for rapid computations [4]. Spark SQL is a Spark module used for structured data processing. It provides native SQL support for the data imported into Spark [5].

Recently, the popularity of NoSQL systems increased, with 3 of the top 11 being NoSQL systems in the DB engines ranking (https://db-engines.com/en/ranking (accessed on 1 May 2022)): MongoDB (5th), Redis (6th), and Cassandra (11th). Consequently, larger enterprises want to shift to NoSQL databases owing to their flexibility, fault tolerance, and availability. However, NoSQL does not use a structured query language (SQL) [6,7] and each model has its own query language. Enterprises face great challenges in converting and migrating RDB to NoSQL because the majority of users are familiar with SQL and SQL is not supported in NoSQL databases [8]. Moreover, the migration to NoSQL has a steep learning curve [9,10]. As the NoSQL model and RDB are complementary, approaches have been proposed to retain the benefits of SQL in the context of NoSQL. However, the previous approaches have two main shortcomings. First, researchers have disregarded a comprehensive approach for executing queries and data manipulation language (DML) on NoSQL models. Second, to the best of our knowledge, there is a lack of research that has conducted experiments in a distributed environment for the migration process to the four models of NoSQL.

These limitations motivated us to introduce a comprehensive Spark-based layer for transforming and migrating RDB to document, column, and key–value databases. The following four significant contributions of this study are presented.

1. Proposing transformation algorithms from the RDB to document, column, and key–value databases.
2. Executing the migration process in a distributed environment using Spark.
3. Executing SQL over document, column, and key–value databases in a distributed environment.
4. Evaluating the proposed Spark-based layer with different scenarios.

The remainder of this paper is organized as follows. Section 2 provides the background of NoSQL models and related works. The proposed layer is described in Section 3. The experiments and results are presented in Section 4. Finally, Section 5 provides concluding remarks.

## 2. Background and Related Work

This section is divided into two parts: NoSQL background and related work. The NoSQL background briefly presents the four models of NoSQL, while related works provide previous studies that have attempted to propose an approach that converts RDB to NoSQL and executes queries and DML on NoSQL.

### 2.1. NoSQL Background

NoSQL models can be classified into two parts: (i) document, column, and key–value and (ii) graph-oriented, which includes only the graph database. Our focus in this study is on the first part.

The document database consisted of collections. Each collection contains several documents. A document contains a set of key–value pairs with a unique key for each document and document content that may be simple or complex. This provides a flexible schema for complex queries. It has high performance and a balanced read/write ratio. MongoDB [11] is an example of a document database.

The column database uses the concept of keyspace. The keyspace comprises column families. Each family contains a set of rows. Each row has a different number of columns. It stores data based on a column distribution schema. This method is suitable for large volumes of data requiring heavy write operations. This supports the use of complex indices and a high aggregation level. Cassandra [12] is an example of a column database.

The key–value database is a simple NoSQL model. It consists of key–value pairs, where the key is the appropriate way to access the value. The value part may be simple or complex, and can be retrieved using an atomic key. The key–value database provides high read/write velocity without frequent updates. This provides a high performance and scalability. Key–value is unsuitable for complex queries that contain joins. Redis [13] is an example of a key–value database.

A graph database consists of nodes and edges, nodes represent entities whereas edges represent the relationship between the nodes. It is used in applications that require traversal between data. It is very suitable for mining data from social media. It is used in many use cases such as fraud detection and recommendation systems. It supports complex query. The most common graph database is Neo4j [14].

*2.2. Related Works*

With the emergence of NoSQL, intermediate layers have been proposed to transform and migrate data from RDB to NoSQL. We briefly describe previous studies, along with the contributions and limitations of each related work.

JackHare [15] is a layer for the transformation and migration from RDB to column data store. HBase was used as a column data store. It uses MapReduce jobs to execute SQL queries on HBase. It supports only a subset of DML and Data Definition language (DDL). It supports SELECT, FROM, WHERE, GROUP BY, HAVING, OREDR BY, JOIN, and aggregate functions using MapReduce. Performance is degraded for a query that involves many join conditions. The main disadvantage of this layer is that it supports only one model from the NoSQL database. It employs the MapReduce processing engine to execute SQL queries, which has a large disc I/O, which slows down the operation.

In MSI [16], the authors propose a layer that allows users to write SQL queries on document and key–value databases. This layer uses Redis as the key–value database and MongoDB as the document database. It uses a nested loop to execute the join operations. It supports a subset of DML and does not support DDL.

Metamorfose [9] is a layer for transformation and migration to document and column data stores. This supports only a subset of DML. This allows the user to customize the mapping. The main disadvantage of this layer is that it uses a small dataset for the experiments. The migration of data is also executed in one node.

Unity [17] is a layer for the transformation and migration to document. It uses MongoDB as a document database. It supports a join using a hash join. This supports only a subset of the DML. Data migration is performed only for a single node. The dataset used in this experiment was small.

SQLtoKeyNoSQL [7,10,18] uses a canonical model for transformation and migration to key–value, document, and column databases. This supports merge and hash join. It supports a subset of DDL and DML. Each SQL is translated into the primitive methods of put, get, and delete. It does not support aggregation or subqueries. The migration step is performed in a single node. However, this does not support the graph database.

SimpleSQL [19] is a layer for transformation and migration to a document database. It uses a simple DB as a document database. It uses a dictionary for mapping from relational to simple DB. It also supports join operations using four steps. The four steps are splitting the query, submitting each split to the interface of simple DB, transforming hits in relational view, and combining steps (joining tables).

In [20], the authors proposed an approach for model transformation and migration from a relational database to a document database. This approach uses the data and query characteristics from the log of the relational database. The model transformation algorithm depends on the description tags. Description tags are frequent join, large size, frequent modification, and frequent insertion. Action tags are generated based on the description tags. Action tags are generated to determine using embedding or references. Automatic data migration was executed after completion of the conversion model. This approach supports DML and join operations.

In [21], a layer that transforms from a relational database to document and column data models was proposed. It allows users to write SQL queries over the MongoDB and Cassandra. This enables SELECT statement other than join operations. It also supports update, delete and insert statements.

BigDimETL [22] is a layer that transforms from a relational database to a column database. MapReduce was used as the processing engine, and HBase was used for flexible

data storage. MapReduce was used in the migration step and query execution. The join was conducted using the MapReduce jobs. The performance was not compared with that of any other layer.

In [23], the authors suggested an approach for transforming a relational database into a document database. It introduces a graph-transformation schema conversion, which captures the tables as nodes and relationships (dependencies) as edges. The main focus of this research is to provide an approach to schema conversion with high performance for join operations using table nesting. The main limitation of this approach is that it requires a large space for query execution.

In [8,24], the authors proposed an approach for transforming relational databases into key–value databases. It uses Oracle NoSQL as a key–value data store. This supports only a subset of DML. It provides data migration validation to ensure the correctness of the data migration.

In NoSQL$^2$ [25], the authors proposed a layer for executing DDL commands in NoSQL databases. It has mainly been proposed for executing administrative tasks. The mapping of DDL was defined manually using a configuration file. This does not support DML and join.

Moratadelo [26] proposed model-driven transformation and migration of relational databases to document and column databases. This provided customization mapping.

In [27], a layer for transformation and migration from RDB to a graph database was proposed. Neo4j was used as a graph database. Spark was used for the migration step. The layer also provides translation from SQL to cypher query language. This supports join and subset of DML.

In [28], the authors proposed a conversion layer to transform and migrate data from RDB to a document database. It uses MongoDB as a document database. The proposed layer chooses a suitable schema for the input RDB based on the application access pattern. The layer defines, evaluates, and compares the candidate NoSQL schemas before the migration process. This supports a subset of DML. It also supports SQL queries that contain joins.

In [29], the authors proposed a layer for converting RDB to a column database. HBase was used as a column database. It uses Apache Phoenix as an SQL layer on top of HBase. This study also discusses techniques and guidelines for migration. The layer supports a subset of DML, join, and subqueries.

In [30], the proposed layer allows users to write SQL queries on top of a document database. It uses MongoDB as a document database. The architecture of the proposed layer has the following modules: (i) database manager, (ii) database adaptor, (iii) general schema driver, and (iv) native driver. To execute SQL queries on MongoDB, the approach generates a general database query with a parameter description, and the general query is translated into the native language of MongoDB.

In [6], the authors proposed a layer that transforms an RDB into a column database. HBase was used as a column database. The layer consists of the following components: (i) an SQL interface for NoSQL, (ii) a database converter, and (iii) a query approach. It uses Apache Phoenix as an SQL translator on top of HBase. Database converter converts data from MySQL to HBase.

To sum up, most of the previous layers support the conversion of RDB to only one or two NoSQL models. In addition, they did not consider parallelization when migrating RDB into NoSQL models, and supported only a subset of select statements. There is an absence of a layer that studies the use of distributed processing engines in migration and query mapping for the models of NoSQL. These layers are summarized in Table 1.

**Table 1.** Comparison between the proposed layers of previous studies.

| Layer | Data Model | Join | Customization Mapping | Automatic Mapping | SQL Support |
|---|---|---|---|---|---|
| JackHare [15] | Column | Supports join with map reduce | No | Yes | DDL + DML subset |
| MSI [16] | Document—key–value | Supports join with nested loop | Yes | No | DML subset |
| Metamorfose [9] | Column—document | - | Yes | No | DML subset |
| Unity [17] | Document | Hash join | No | Yes | DML subset |
| SQLTOKEYNoSQL [7,10,18] | Column—document—key–value | Merge join and hash join | No | Yes | DDL + DML subset |
| SimpleSQL [19] | Document | Supports join by similarity | No | Yes | DDL + DML subset |
| [20] | Document | Support join | Yes | No | DML subset |
| [21] | Document—column | - | No | Yes | DML subset |
| BigDimETL [22] | Column | Supports join by using MapReduce paradigm | No | Yes | DML subset |
| [23] | Document | Supports join | No | Yes | DML subset |
| [8,24] | Key–value | Supports join | No | Yes | DML subset |
| NoSQL² [25] | Four models | - | Yes | No | Only DDL |
| Moratadelo [26] | Column—document | - | Yes | No | - |
| [27] | Graph database | Supports join | No | Yes | Subset of DML |
| [28] | Document | Supports join | Yes | No | Subset of DML |
| [29] | Column | Supports join | No | Yes | Subset of DML |
| [30] | Document | Supports join | No | Yes | Subset of DML |
| [6] | Column | Supports join | No | Yes | Subset of DML and DDL |

Thus, this research proposes a comprehensive Spark-based layer that covers the following: First, converts the relational database into document, column, and key–value databases.

Second, executes SQL queries that contain all types of joins, subqueries, and aggregate functions on converted NoSQL databases in a distributed environment.

Third, translates SQL insert, update, and delete statements into their equivalents in NoSQL database.

## 3. The Proposed Architecture of the Spark-Based Layer

The proposed Spark-based layer consists of two parts: (i) converting the relational database to NoSQL databases, and (ii) executing SQL on NoSQL. Figure 1 shows the architecture of the proposed Spark-based layer.

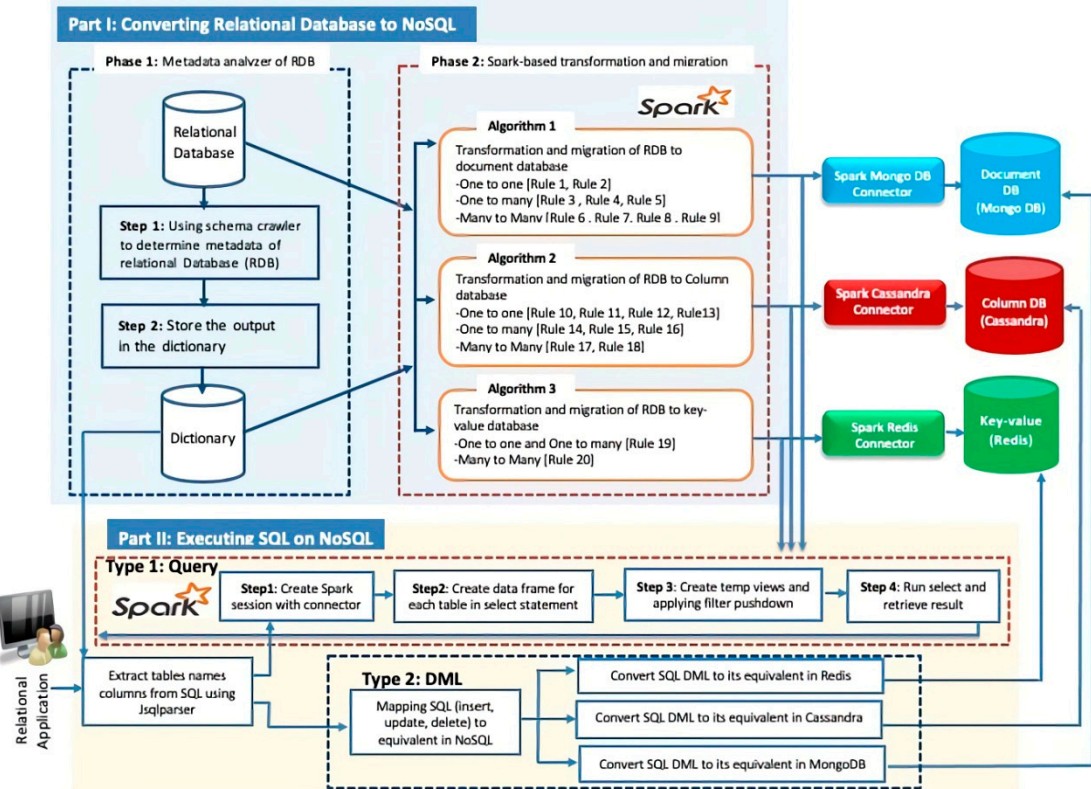

**Figure 1.** The architecture of the proposed Spark-based layer (created by the author).

### 3.1. Part I: Converting the Relational Database to NoSQL

This subsection converts the traditional database into a NoSQL database. The process of conversion includes two phases, which are explained in Sections 3.1.1 and 3.1.2, respectively.

#### 3.1.1. Phase 1: Metadata Analyzer of the RDB

Phase 1 consists of two steps: Step 1 is a schema metadata analyzer and Step 2 stores the output in the dictionary. A schema metadata analyzer is responsible for extracting the metadata. A schema metadata analyzer uses a schema crawler [31] to extract metadata from the RDB. It connects to the RDB and extracts the table's names, fields, primary keys, foreign keys, and relationships between the tables. In Step 2, the metadata extracted in Step 1 are stored in a dictionary. The dictionary maintains metadata for each RDB table (fields, primary keys, and foreign keys). It also retains the one-to-one, one-to-many, and many-to-many relationships for a given RDB.

#### 3.1.2. Phase 2: Spark-Based Transformation and Migration

The main objective of this phase is to transform and migrate the RDB to the target NoSQL database. This phase encompasses the following three steps.

1. Algorithm 1: Transforms and migrates to a document database.
2. Algorithm 2: Transforms and migrates to a column database.
3. Algorithm 3: Transforms and migrates to a key–value database.

The inputs to this phase were the RDB and metadata provided by the dictionary. The proposed algorithms rely on Spark and the connectors between Spark and NoSQL databases as part of the migration process.

Algorithm 1: Transformation and Migration from RDB to Document

Document stores consist of collections, and each collection contains documents stored in it. Table 2 depicts the mapping of concepts between the relational and document databases. Some scholars have used the following strategy: (i) each table is migrated as

a collection and (ii) each row is mapped as a document [7,19]. However, the proposed layer uses Spark as the processing engine for the migration process (in a distributed environment). The transformation and migration algorithm from an RDB to a document database is presented in Algorithm 1.

**Table 2.** Mapping concepts between relational databases and document databases.

| Entity Relationship Model | Relational Database | Document Database |
|---|---|---|
| Entity | Table | Collection |
| Relationships | Constraint | Reference or embedding or bucketing |
| Attribute | Column | Field |
| Entity instance | Row | Document |

Algorithm 1 provides two modes for the transformation and migration to document databases: (i) automatic transformation, and (ii) transformation based on user selection rules. Automatic transformation uses a dictionary to analyze the RDB and construct a suitable schema for a given RDB. However, in a transformation based on user selection rules, the user selects specific rules to convert each relationship in the RDB. For example, automatic transformation applies Rule 1 embedding one-to-one if the referencing table does not have a relationship with other tables in the RDB; however, the user may select Rule 2 in this case. This algorithm contains nine rules for transformation from an RDB to a document database, which are classified as follows:

One-to-one relationships were mapped using an embedded strategy [20] or referencing [32]. The rules for mapping one-to-one are as follows:

1.  Rule 1, embedding one-to-one: If the referencing table does not have a relationship with other tables in the RDB, embed referencing objects into the referenced. The primary key of the referenced table is converted as the key of the collection. There is no need to store the field that represents the foreign key because the referencing table and referenced table are migrated to the same collection in the document database.
2.  Rule 2, referencing one-to-one: If the referencing table has a relationship with other tables in the RDB, add referencing object references to referenced objects (represents foreign key conversion). The primary key of the referenced table is converted as key of the collection and the primary key of the referencing table is converted as key of the collection.

One-to-many relationships can be modelled by embedding or referencing [33]. The rules for mapping one-to-many are as follows:

1.  Rule 3, embedding one-to-many: Embedding the many-side on the one-side if the many-side are not referenced by other tables [33].The primary key of the one-side table is converted as the key of the collection. The many-side is converted as an array inside the one-side, so no need to store the field that represents the foreign key.
2.  Rule 4, referencing one-to-many, one-side: If the one-side of the one-to-many relationship has relationships with other tables or a small number of the many-side is associated with the one-side, it is transformed by adding into the one-side an array of references of the many-side [33].
3.  Rule 5, referencing one-to-many, many-side: If the many-side of the one-to-many relationship has relationships with other tables or a large number of the many-side is associated with the one-side, it is transformed by adding a many-side reference to the one-side [10].

Many-to-many relationships can also be modelled by referencing or embedding. The selection of embedding or referencing is based on the system requirements. The rules for mapping many-to-many are as follows:

1.  Rule 6, two-way embedding many-to-many N:M: If the size of M is close to N [32].

2. Rule 7, one-way embedding many-to-many N:M: If the size of M is significantly greater than N.
3. Rule 8, creates a document collection that represents the M:N relationship: Creates document collection for each participating table in M:N, and creates a document collection that includes the columns of the M:N relationship and foreign keys for participating tables in the relationship [7,10].
4. Rule 9, referencing to M:N using references to avoid data redundancy [32–34].

The recursive (self) relationship in RDB is a foreign key relationship in the same table. For example, employee has empId as a primary key and "reportsTo" as a foreign key that represents the recursive relationship. The "reportsTo" column in the RDB is migrated as a field in the document database. Figure 2 shows an example of a simple relational RDB. Figure 3 shows the document database model after applying rules 2, 5, and 8.

The **ConstructAutomaticSchema** function is used to automatically construct a document database model, whereas **ConstructUserSelectedSchema** is used to create schemas by applying the rules given by the user.

The ConstructUserSelectedSchema is pretty similar to this work [35]. This work proposes an automatic suggestion model based on the user defined system requirements and CRUD (Create, Read, Update, Delete) operations.

---

**Algorithm 1:** TRANSFORMATION AND MIGRATION FROM AN RDB TO A DOCUMENT MODEL

---

|   | **Input: RDB r, Dictionary, User selected rules** |
|---|---|
|   | **Output**: RDB r transformed and migrated to document database d |
| 1 | **DocumentDatabase d** ← create Document Database |
| 2 | **OneToOne []** ←Extracting One-To-One Relationships From Dictionary of given RDB |
|   | // OneToOne [] contains referencing table and referenced table of each one-to-one relationship |
| 3 | **OneToMany []** ←Extracting One-To-Many Relationships From Dictionary of given RDB |
|   | // OneToMany [] contains referencing table and referenced table of each one-to-Many relationship |
| 4 | **ManyToMany []** ←Extracting Many-To-Many Relationships From Dictionary of given RDB |
|   | /* ManyToMany [] contains metadata of two participating tables that is connected by join table of each M:N relationship and bridge table (join table) */ |
| 5 | **If** (automatic selection) **//** |
| 6 |    **Function:** ConstructAutomaticSchema (r,d, OneToOne [],OneToMany [], ManyToMany []) |
| 7 | **Else** |
| 8 |    **Function:** ConstructUserSelectedSchema (r,d, OneToOne [],OneToMany [], ManyToMany []) |

---

The **frequentjoin** () function is used to check whether the table is frequently queried together using a join. The **getNumberOfRelationships** () method is used to obtain the number of relationships in this table.

---

**Function**: ConstructUserSelectedSchema (r, d, OneToOne [],OneToMany [], ManyToMany []))

---

| 1 | **UserSelectedRules []** ←GetUserSelectedRules () // user manually select each rule |
|---|---|
| 2 | **For each** rule: UserSelectedRules [] |
| 3 | **If rule** ∈ *{Rule 1, Rule 2}* |
|   |    **TransformAndMigrateOneToOne (rule, OneToOne [])** |
| 4 |    /* TransformAndMigrateOneToOne method takes a user-selected rule of one-to-one relationship and migrates data to document database using Spark */ |
| 5 | **Else If** rule ∈ *{Rule 3, Rule 4, Rule 5}* |
|   |    **TransformAndMigrateOneToMany (rule, OneToMany [])** |
| 6 |    /* TransformAndMigrateOneToMany method takes the user selected rule of one to many relationships and migrates data to document database using Spark */ |
| 7 | **Else** |
|   |    **TransformAndMigrateManyToMany (rule, ManyToMany [])** |
| 8 |    /* TransformAndMigrateManyToMany method takes the user-selected rule (rule 6 or rule 7 or rule 8 or rule) of many-to-many relationships and migrates data to document by using Spark */ |
| 9 | **End for each** |



| **Function**: ConstructAutomaticSchema (r,d, OneToOne [], OneToMany [], ManyToMany []) |
|---|

| | |
|---|---|
| 1 | **For each** relationship e: OneToOne [] |
| 2 | **Table t** ←e.getReferencingTable () |
| 3 | **If** (t.getChildTable () = = null) |
| 4 | Apply rule 1 Embedding one-to-one |
| 5 | **Else** |
| 6 | Apply rule 2 Referencing one-to-one |
| 7 | **End for each** |
| 8 | **For each** relationship e: OneToMany [] |
| 9 | **Table t1** ←e.getReferencingTable () |
| 10 | **Table t2** ←e.getReferencedTable () |
| 11 | **If** (t1.getChildTable () = = null) |
| 12 | Apply rule 3 Embedding one to many |
| 13 | **Else If** ((t1.getNumberOfRelationships () > t2.getNumberOfRelationships ())) |
| 14 | Apply rule 4 referencing one to many one-side |
| 15 | **Else** |
| 16 | Apply rule 5 referencing one to many many-side |
| 17 | **End for each** |
| 18 | **For each** relationship e: ManyToMany [] |
| 19 | **Table t1, t2** ←e.getTables () |
| 20 | **If** frequentjoin (t1, t2) == true // if two participated tables in M:N are frequently join |
| 21 | **If** (t1.size ()—t2.size () <= threshold) **//** if t1 size is close to t2 size |
| 22 | Apply rule 6 two way embedding many to many N:M |
| 23 | **Else** |
| 24 | Apply rule 7 one way embedding many to many N:M |
| 25 | **Else** |
| 26 | Apply rule 9 referencing M:N |
| 27 | **End for each** |

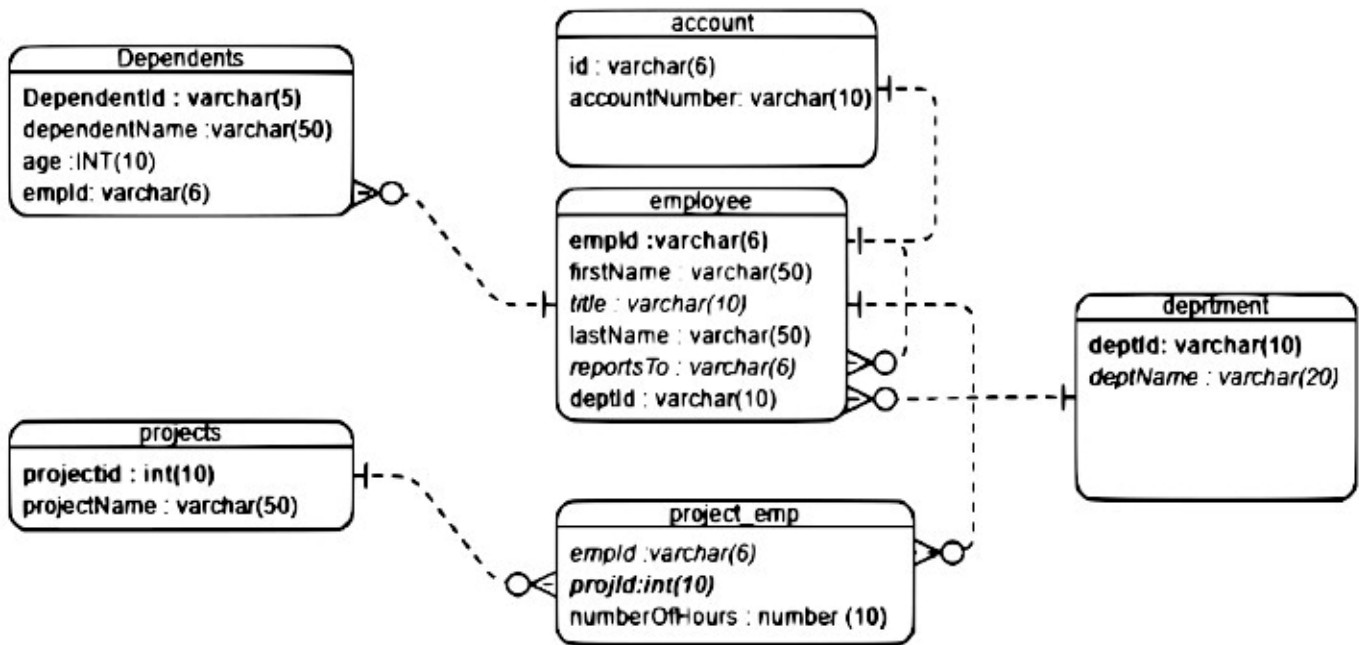

**Figure 2.** Simple RDB example.

**Dependents**

"_id": " Dep01",
"dependentId":"Dep01",
"dependentName": "Kasim",
"age":"10",
"empId":"E012"

**Department**

"_id": " Dept02",
"deptId":"Dept02",
"deptName": "manufacturing"

**Employee**

"_id": " E012",
"empId":"E012",
"title": "Eng",
"firstName":"Tariq",
"lastName":"zeiad",
"deptId":"Dept02" ,
"reportsTo":"E011"

**Project**

"_id": "12",
"projectId":"12",
"projectName":"NewCapital"

**Project_Emp**

"_id": " E012.12",
"projectId":"12",
"empId":" E012",
"numberOfHours":"10"

**Account**

"_id": " ac01",
"id":"E012",
"accountNumber":"ac01"

**Figure 3.** Document schema generated by applying rules 2, 5 and 8 to Figure 2.

Algorithm 2: Transformation and Migration from RDB to Column

Converting an RDB into a column store depends on data access patterns. Frequently accessed data should be stored within the same column family as much as possible. If the proposed schema is not suitable for the access pattern after applying the conversion, it will lead to redesigning the schema to fulfill accessibility requirements [36]. Algorithm 2 shows the transformation and migration from an RDB to a column database. The algorithm contains nine rules for the transformation from the RDB to the column database, and is classified as follows:

A one-to-one relationship in the RDB can be converted into a column database by using the following rules:

(1) Create one table: this pattern is classified to different strategies

    (a) Rule 10 (single family): Merge all fields of the two RDB tables into a single column family in a column store database [37].

    (b) Rule 11 (two-column family): A single table consists of one column family for the first table and another column family for the second table [37].

(2) Creating two tables:

    (a) Rule 12, creates a column family for each participating table in the RDB, where the row key of the column family is the primary key in the RDB table and the primary key of the one-side becomes a column on the many-side. [38].

    (b) Rule 13, creates a column family for each participating table in the RDB, where each column family contains all fields of the two tables [37].

One-to-many relationships in the RDB can be converted into a column database by using the following rules:

1. Rule 14 (creating one table with a single-column family in the column store database): Merge all fields of the two tables in the RDB to a single-column family [37].

2. Rule 15 (creating one table with a two-column family): Create a super column family in which each table is stored in a separate column family, where the table on the one-side contains a table on the many-side and the row key is the primary key of the one-side [38].
3. Rule 16 (creating two tables with one column family per RDB table): Each table in the RDB is represented as a table in the column store, where the table in the column store has a single column family. The row key is the primary key of RDB table, as in this work [7] and the other layers concatenate the RDB table name with the primary key value to generate the row key [15,39].

Many-to-many relationships in the RDB were converted into a column database using the following rules:

1. Rule 17 (create three tables with one column family per RDB table): Each table in the many-to-many relationship is mapped to a table in a column store, where the row key of the bridge RDB table is a concatenation of the composed primary key [7].
2. Rule 18 (create a column family for each table and a super column family): Create a column family for each table in the relationship, where the row key of the column family is the primary key of the RDB table. Create a super column family for the bridge RDB and include all columns of the two RDB tables. The row key of the super family is the concatenation between the relationship name and the primary key of one column that is frequently used in the query [38].

The column that represents the recursive relationship can be converted to a field in the column database such as "reportsTo" in the employee table in Figure 4. Figure 4 presents the column store after applying rule 12 for one-to-one relationships, rule 16 for one-to-many relationships, and rule 17 for many-to-many relationships.

| **Algorithm 2**: TRANSFORMATION AND MIGRATION FROM AN RDB TO A COLUMN MODEL | |
|---|---|
| | **Input**: RDB r, user selected rules, dictionary |
| | **Output**: RDB r transformed and migrated to column database d |
| 1 | **Keyspace d** ← createKeyspace (r) |
| 2 | **OneToOne []** ←Extracting One-To-One Relationships From Dictionary of given RDB |
| 3 | **OneToMany []** ←Extracting One-To-Many Relationships From Dictionary of given RDB |
| | **ManyToMany []** ←Extracting Many-To-Many Relationships From Dictionary of given RDB |
| 4 | /* ManyToMany [] contains metadata of two participating tables that is connected by join table of each M:N relationship and bridge table (join table) */ |
| 5 | **UserSelectedRules []** ←GetUserSelectedRules () |
| 6 | **For each** rule: UserSelectedRules [] |
| 7 | **If rule** ∈ *{Rule 10, Rule 11, Rule 12, Rule 13}* |
| | **TransformAndMigrateOneToOneCoulmn (rule, OneToOne [])** |
| 8 | /* TransformAndMigrateOneToOneCoulmn method takes the user-selected rule of a one-to-one relationship and migrates data to the keyspace using Spark */ |
| 9 | **Else If rule** ∈ *{Rule 14, Rule 15, Rule 16}* |
| | **TransformAndMigrateOneToManyCoulmn (rule, OneToMany [])** |
| 10 | /* TransformAndMigrateOneToManyCoulmn method takes a user-selected rule of one to many relationships and migrates data to the keyspace using Spark */ |
| 11 | **Else** |
| | **TransformAndMigrateManyToManyCoulmn (rule, ManyToMany [])** |
| 12 | /* TransformAndMigrateManyToManyCoulmn method takes a user-selected rule of many-to-many (rule 17 or rule 18) relationships and migrates data to the keyspace by using Spark */ |
| 13 | **End for each** |

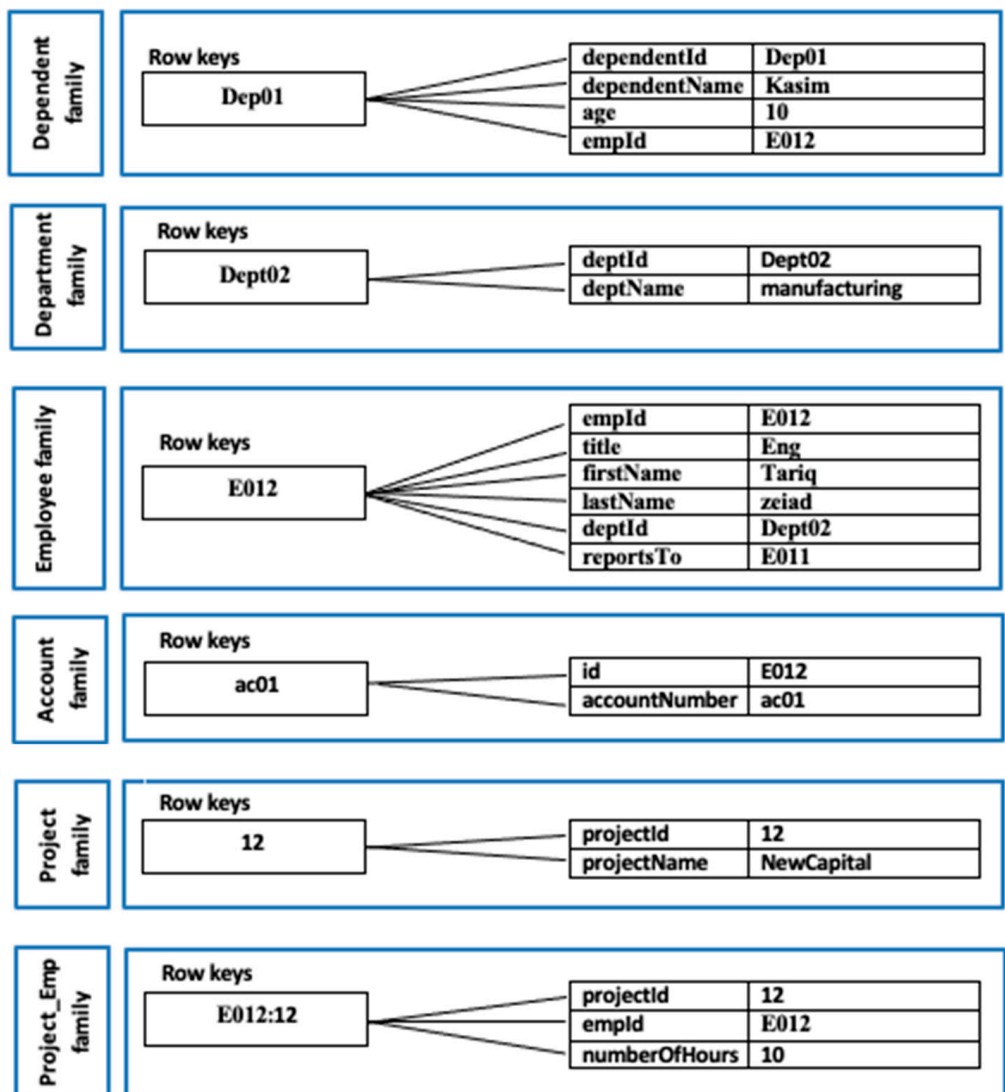

**Figure 4.** Column schema generated by applying rules 12, 16, and 17 to Figure 2.

Algorithm 3: Transformation and Migration from RDB to Key–Value

The transformation from RDB to key–value database is handled as follows:

One-to-one and one-to-many relationships in the RDB can be converted into a key–value database by using the following rule:

Rule 19: The key in the key–value store is the concatenation of the relational table name and value of the primary key of this table. The value in the key–value store is a set of key–value pairs that consists of fields with their values in the relational table.

Many-to-many relationships in the RDB can be converted into a key–value database by using the following rule:

Rule 20: In the case of migrating a bridge table to a key–value store, the key is the concatenation of the relational table name and the values of the columns in the composed primary key, such as project_emp:12. E012 in Figure 5. The value in the key–value store is a set of key–value pairs that consists of fields with their values in the relational table.

For example, Project_emp is the relational table name, 12 is the value of Project Id (part of the composed primary key), and E012 is the value of empId (part of the composed primary key). Figure 5 presents the key–value store after applying the transformation rules. Algorithm 3 shows the transformation and migration of the key–value database.

| | |
|---|---|
| **Algorithm 3**: TRANSFORMATION AND MIGRATION FROM AN RDB TO KEY–VALUE MODEL | |

| | |
|---|---|
| | **Input**: RDB r and dictionary |
| | **Output**: RDB r transformed and migrated to key-value database d |
| 1 | **OneToOne []** ←ExtractingOneToOneRelationshipsFromDictionary of given RDB |
| 2 | **OneToMany []** ←ExtractingOneToManyRelationshipsFromDictionary of given RDB |
| 3 | **ManyToMany []** ←ExtractingManyToManyRelationshipsFromDictionary of given RDB <br> // ManyToMany [] contains the bridge table with its columns only |
| 4 | **For each** table t ∈ {OneToOne [], OneToMany []} <br> // for each table participating in one-to-one or one-to-many relationship |
| 5 | **DataFrame df** ← ReadRelationalDataUsingSpark (t, r) |
| 6 | **tableName**←t.getName () |
| 7 | **Pk**← t.getPK () // get primary key of table |
| 8 | **For each** element e in df |
| 9 | **Key** ← tableName+ ":"+ pk.getColumnValue ()**// construct key by concatenating the table name and the value** <br> of primary key column |
| 10 | **Value**← e.getColumnNamesAndValues ()// construct value by generating a set of key-value pairs that <br> represent the fields with their values |
| 11 | d←Emits(Key, Value)**// store the key and key-value pairs in key-value database** |
| 12 | **End for each** |
| 13 | **End for each** |
| 14 | **For each** table t ∈ ManyToMany [] <br> **//** for each bridge table in Many-To-Many |
| 15 | **DataFrame df** ← ReadRelationalDataUsingSpark (t, r) |
| 16 | **tableName**←t.getName () |
| 17 | **Pks []**← t.getPKs () **//** get columns in the composed primary key |
| 18 | **For each** element e in df |
| 19 | **Key** ← tableName + ": "+ pks.getPKsValueConcatenated () <br> // pks.getPKsValueConcatenated () get the values of the columns in composed primary key <br> // construct key by concatenating the table name and the values of primary key columns |
| 20 | **Value**← e.getColumnNamesAndValues () <br> **//** construct value by generating a set of key-value pairs that represent the fields with their values |
| 21 | d←Emits(Key, Value) **//** store the key and key-value pairs in key-value database |
| 22 | **End for each** |
| 23 | **End for each** |

The column that represents the recursive relationship can be converted as a field in the key–value database such as "reportsTo" in employee table in Figure 5.

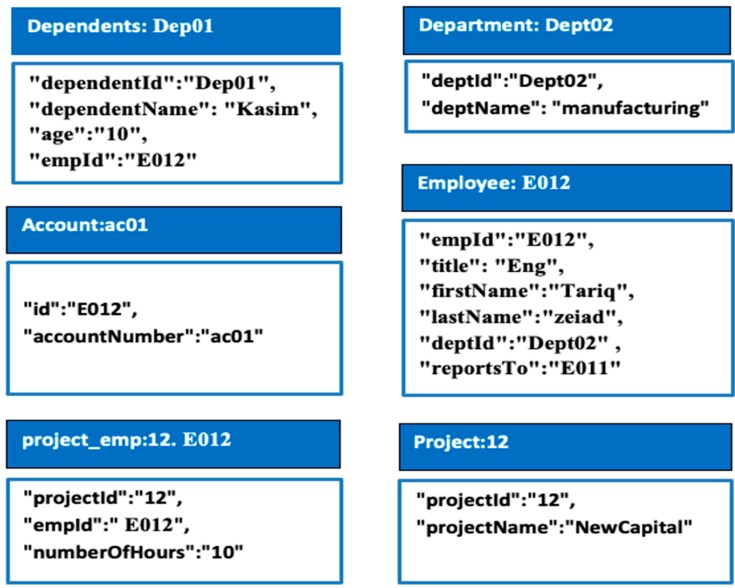

**Figure 5.** Key–value schema generated by applying rules 19 and 20 to Figure 2.

*3.2. Part II: Executing SQL on NoSQL*

In Section 3.2, we discuss the addition of SQL features to NoSQL databases, as illustrated in Figure 1. This subsection is divided into two types.

- Type 1 (Query): Executing SELECT statements (explained in Section 3.2.1)
- Type 2 (DML): Converting INSERT, UPDATE, and DELETE to their equivalents in the NoSQL database (explained in Section 3.2.2)

Initially, Jsqlparser is used to extract table names, column names, and values when executing SQL commands [40]. The execution of SQL commands uses the metadata maintained by the dictionary.

### 3.2.1. Type 1: Query

The execution of SQL queries on a NoSQL database depends on Spark SQL. Spark SQL is a structured data processing module [5]. It can be used to read data from external databases using a DataFrame interface. This allows users to write SQL over NoSQL databases by registering the data frame in a temporary view. It also has an automatic schema inference. Additionally, it can be used as a distributed query engine.

The proposed layer uses Spark SQL because it supports all types of joins, subqueries, and aggregation functions. Therefore, we integrated Spark SQL into the proposed layer. The proposed layer uses connectors to integrate Spark with NoSQL databases. We used the Spark MongoDB connector to integrate Spark SQL with MongoDB [41], the Spark Redis connector [42] to integrate Spark SQL with Redis, and the Spark Cassandra connector [43] to integrate Spark SQL with Cassandra. The major difference between executing SELECT in the proposed layer and the layers in the literature survey is that the proposed layer executes SELECT queries in a distributed environment. The layers in the literature survey support only a subset of the SELECT queries.

Filter pushdown (predicate pushdown) is a technique used to improve the query performance. The filter pushdown reduces the amount of data transferred from data storage, such as NoSQL databases, to Spark worker nodes. It executes filters, such as query conditions (WHERE clause), in SELECT at the NoSQL data source before loading the data to the Spark worker nodes. Therefore, the time required to execute a query is reduced.

The following steps are applied to execute SELECT queries on NoSQL databases:

- **Step 1**: Create a Spark session that connects Spark with NoSQL.
- **Step 2**: Create data frames for each table in the SELECT statement by using Spark SQL and connectors.
- **Step 3:** Create temporary views from DataFrames and apply filter pushdown if the query contains filter conditions such as WHERE clause.
- **Step 4:** Run the SELECT statement on temporary views and retrieve the result as a DataFrame for relational application.

### 3.2.2. Type 2: DML

The SQL DML type translates INSERT, UPDATE, and DELETE statements into their equivalents in NoSQL stores. It uses the output from Jsqlparser and the metadata of the RDB stored in the dictionary. For example, the following insert statement "**insert into actors values ('ahmed', 'mohamed', 'm')**" does not have column names. Therefore, we must obtain the names of columns from the data maintained in the dictionary. The layer maps a subset of SQL DML statement commands to their equivalents and then executes these commands on MongoDB, Cassandra, and Redis databases.

### Converting SQL DML to MongoDB

For example, the insert statement "insert into actors (first_name, last_name, gender) values ('ahmed', 'mohamed', 'm')" is translated to "db.actors.insertOne({first_name: 'ahmed', last_name: 'mohamed', gender: 'm'})".

To translate the SQL DELETE into its equivalent in the MongoDB, the layer extracts the table name and filter conditions from the deleted statement. For example, the delete statement "**delete from actors where id = 3**" is translated automatically as "**db.actors.deleteMany({id:3}**)".

This layer provides a translation of the SQL UPDATE to **updateMany()** in the MongoDB command by extracting the table name and filter conditions from the UPDATE statement. Table 3 provides a sample translation from SQL to MongoDB.

**Table 3.** Translate UPDATE statements from SQL to MongoDB the using proposed layer.

| SQL Statement | MongoDB |
|---|---|
| update actors SET gender= 'f', last_name = 'gamal' WHERE id <= 3 | db.actors.updateMany({id: {$lte: 3}}, {$set: {gender: 'f', last_name: 'gamal'}}) |
| update actors SET gender= 'f', last_name = 'gamal' WHERE id = 3 | db.actors.updateMany({id: {$eq: 3}}, {$set: {gender: 'f', last_name: 'gamal'}}) |
| update actors SET gender= 'f', last_name = 'gamal' WHERE id > 3 | db.actors.updateMany({id: {$gt: 3}}, {$set: {gender: 'f', last_name: 'gamal'}}) |

Converting SQL DML to Cassandra

UPDATE, INSERT, and DELETE statements in Cassandra are very similar to SQL.

Converting SQL DML to Redis

An example of the translation from the insert to Redis is presented by St1 in Table 4. The layer provides translation from the SQL UPDATE to the **hmset** command, as shown by St2 in Table 4. An example of a translation from the SQL DELETE statement is presented by St3 of Table 4.

**Table 4.** Translation from SQL to Redis using the proposed layer.

| Stat. # | SQL Statement | Redis |
|---|---|---|
| St1 | insert into actors (first_name, last_name, gender)values('ahmed', 'kazim', 'm') | hmset actors:first_name 'ahmed' last_name 'kazim' gender 'm' |
| St2 | update actors SET gender= 'f', last_name = 'gamal' WHERE id = 845467 | hmset actors:845467 gender 'f' last_name 'gamal' |
| St3 | delete from actors where id = 3 | DEL actors:3 |

## 4. Results and Discussion

In this section, the performance of the proposed Spark-based layer is evaluated. Section 4.1 outlines the databases used in the experiments and their characteristics. Section 4.2 presents the experiments and results.

### 4.1. Databases

The IMDB [44] is an open-source database. It is available in the form of plain-text files. Only a subset of the database was used for the experiments. Table 5 presents two databases with their characteristics, IMDB and Stack Exchange. For the migration evaluation, the data stored in each table in the IMDB were copied to a new table with different names to obtain size 2. Each table in the IMDB is copied to two new tables with different names to get size 3. Size 1 of the IMDB is 5,610,922 tuples, size 2 is 11,221,844 tuples, and size 3 is 16,832,766 tuples.

The Stack Exchange dataset [45] is a benchmark dataset used in the experiments. We used only the Users and Votes tables from the dataset. The Votes table is uncompressed by 20 GB with 222 million tuples, whereas the Users table is uncompressed by 5 GB with more than 13 million tuples, which means that the Stack Exchange dataset contains more than 235 million tuples. The Stack Exchange dataset is in the XML format. We constructed

a big data pipeline using Spark to load the dataset into the MySQL. The XML dataset was converted into a CSV file, which was then divided into smaller CSV files to speed up data loading by increasing the level of parallelism to MySQL, as illustrated in Figure 6.

**Table 5.** Characteristics of the databases.

| Databases | Number of Tables | Tuples | Relationships |
|-----------|------------------|--------|---------------|
| IMDB | 7 | 5,610,922 | 6 |
| Stack Exchange | 2 | 235,000,000 | 1 |

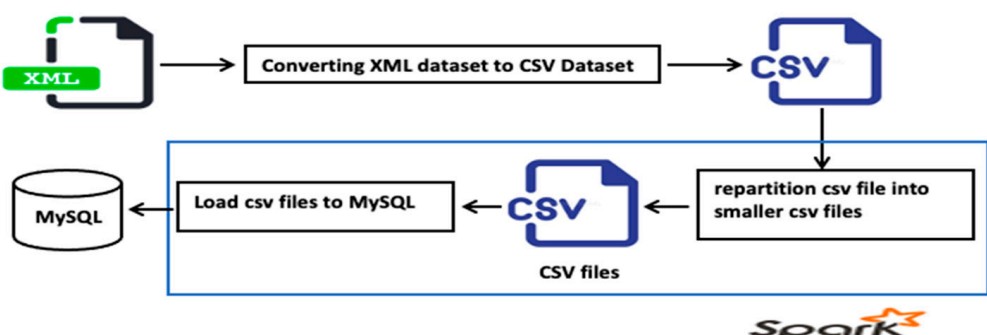

**Figure 6.** The pipeline for preprocessing of the Stack Exchange dataset.

*4.2. Experiments and Results*

The proposed layer was implemented using Java programming language and Apache Spark 2.4.1. MySQL has been used as a relational data source, MongoDB 4.3 as a document database, Cassandra 3.11 as a column database, and Redis as a key–value database. Experiments were conducted on a Spark standalone cluster with four nodes. One node serves as the standalone master for Spark and three nodes serve as Spark worker nodes. Each node had an Intel Core i7 processor and 8 GB RAM. The main objective of this research is not to test the performance of MongoDB vs. Cassandra vs. Redis but to evaluate write (migrate) and read (SQL select) using the proposed Spark-based layer. The results may vary when experiments are conducted on a hard disk drive (HDD) versus a solid-state drive (SSD). All nodes were connected using a switch of 10/100 MBs. All the experiments in this study were performed on an HDD hard disk.

To improve the ingestion of data from a relational database into the Spark cluster, we tuned the following properties:

1. Number of partitions: maximum number of partitions that can be used for table reading. This property also determines the number of concurrent connections between Spark workers and the relational database because, by default, only one executor in the Spark cluster works.
2. Partition column: the name of the column used for partitioning the table.
3. Lower bound and upper bound: control the partition stride.

4.2.1. The Proposed Layer vs. Unity in a Single-Node Environment

The proposed layer is evaluated in this subsection. A qualitative comparison between the proposed layer and the Unity model [17] is presented in Table 6. Unity is an SQL interface to access relational and document databases. It allows the execution of SQL queries that contain join conditions, WHERE, GROUP BY, and ORDER BY on the NoSQL database. It consists of parser, translator, optimizer, and execution engine. The parser transforms SQL queries into a parsed tree. The translator converts the parsed tree into a relational operator tree. The optimizer determines the join ordering. The execution engine interacts with the NoSQL data source and submits the queries to the NoSQL data source via APIs and receives the results. Unity uses MongoDB as the NoSQL data source.

**Table 6.** Qualitative comparison between the proposed layer and Unity.

| Features | Unity | Proposed Layer |
| --- | --- | --- |
| Access document data store | Yes | Yes |
| Access column data store | No | Yes |
| Access key–value data store | No | Yes |
| Supports join operation | Yes | Yes |
| Supports aggregate functions | Yes | Yes |
| Supports subquery | Yes | Yes |
| Transformation and migration using distributed processing engine | No | Yes |

The IMDB was migrated from MySQL to the MongoDB using the proposed layer. The comparison between Unity and the proposed layer in the MongoDB database for executing the following SQL query "**select d.first_name, m.name from directors d, movies m movies_directors md, roles r where d.did = md.director_id and md.movie_id = m.mid and m.mid = r.movie_id**" is depicted in Figure 7. For size 1 and size 2, the Unity framework outperformed the proposed layer. However, with a database of size 3, the proposed layer outperformed Unity. This is because the Spark MongoDB connector creates multiple concurrent connection pools in the MongoDB, where each incoming connection receives one thread and each parallel-processing thread is run in a separate core. However, Unity only uses a single connection pool. Therefore, the proposed layer outperformed Unity in terms of the query execution time by a factor of three.

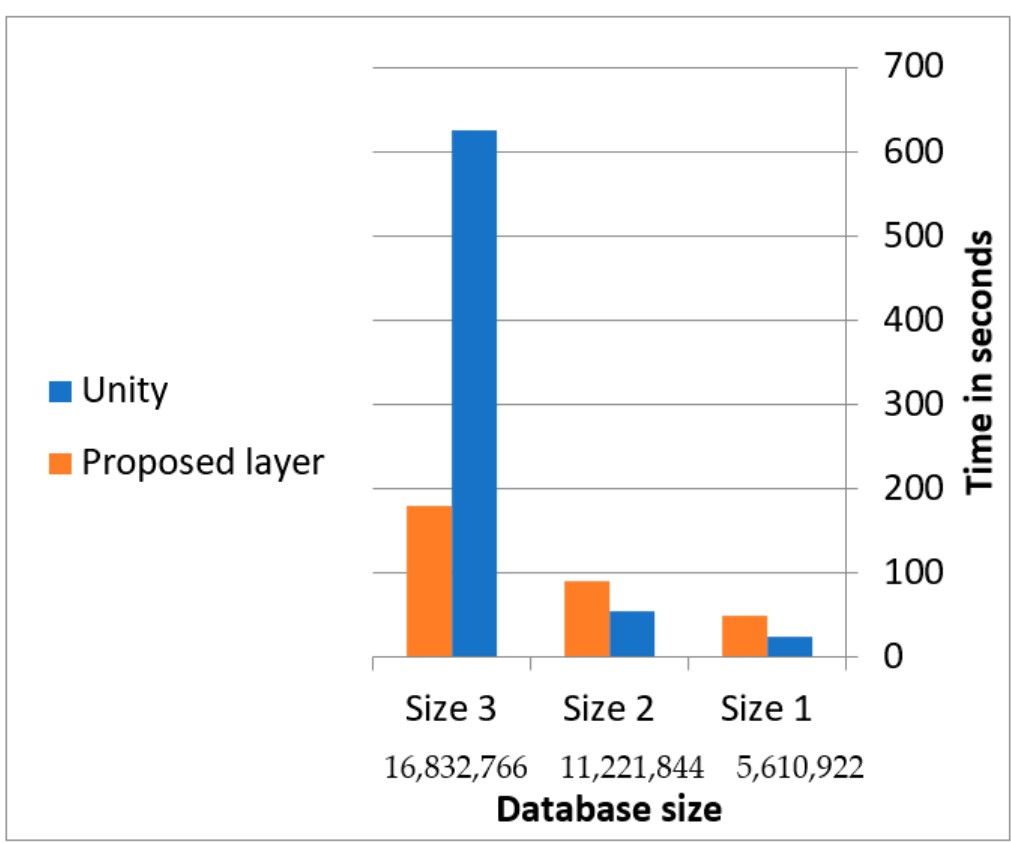

**Figure 7.** Elapsed time comparison between Unity and the proposed layer.

### 4.2.2. Document Database Evaluation

We also created MongoDB shared clusters to evaluate the performance of our proposed layer. The sharded Mongo DB cluster consisted of the following components: (i) the config server, which stores the metadata and configuration settings for the sharded cluster; (ii) mongos, the query router that acts as an interface between the client application and the sharded cluster; and (iii) a shard, each containing a subset of the sharded data. To achieve data locality, a sharding MongoDB cluster was installed on the same machine as the Spark cluster. Each node in the Spark standalone cluster has a processor with eight cores and 8 GB of RAM. Figure 8 displays the elapsed time for the migration of data using the Spark local mode, Spark standalone mode, and sharding and non-sharding MongoDB. As shown in Figure 8, for size 3, the Spark multi-node cluster with sharding database outperformed the Spark multi-node cluster with no sharding database by 50%, the Spark local with sharding database by 16%, and the no sharding database by 36%.

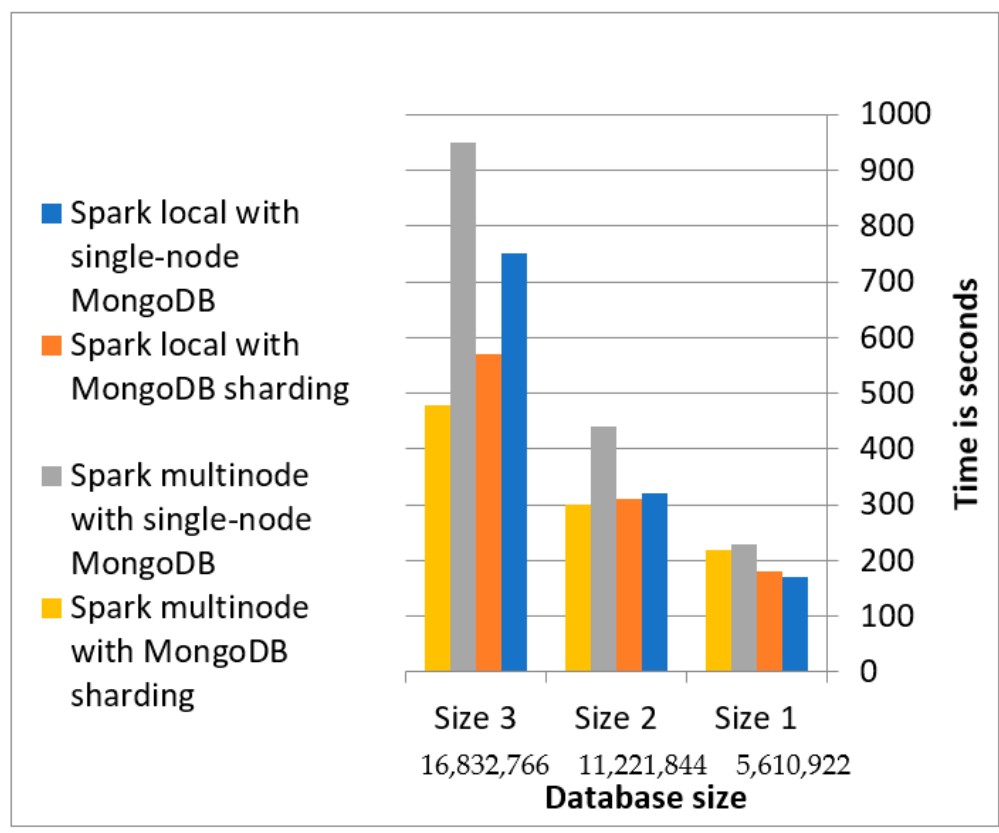

**Figure 8.** Elapsed time for migrating data to MongoDB using the proposed layer.

The migration time is improved in the Spark multi-node with MongoDB sharding because the sharding cluster can balance the incoming concurrent connections from the spark multi-node cluster. Therefore, Spark local (multiple concurrent connections) with sharding is better than Spark multi-nodes with no sharding. In the case of a Spark multi-node with a sharded database scenario, the parallelizability of the writes is increased by applying the following:

1.  Inserting to a sharded collection.
2.  The indexes are removed at the time of writing data to sharpen the collection and reconstruct the indexes after the end of the writing operation.
3.  Setting writes concerns only the primary node.
4.  Ensuring data locality.

To evaluate the performance of the MongoDB query, we used the same query to compare Unity and our proposed layer. To speed up the query execution time, a shared key

was constructed for each collection during the migration step. Each table in the migration step is migrated to a collection, in which the shard key is the primary key of the tables. The shard key of the bridge table is a compound shard key that has all attributes of the composite primary key. When an application driver issues a query that includes a shard key to the mongos (query router server), the mongos uses the metadata from the config server to route the query to the shards. The Spark Mongo connector uses a predicate pushdown that executes filters at the data source (MongoDB) before it is loaded into Spark. It is important to carefully design a shared cluster based on each use case. Figure 9 shows the elapsed time required to execute queries in different scenarios. When executing queries for a database with size 3, the Spark multi-node cluster with sharding database outperforms the Spark multi-node cluster with no sharding, Spark local with sharding, and Spark local with no sharding by 30%, 25%, and 34%, respectively, because sharding the database opens multiple concurrent connections from Spark local mode or Spark multi-node.

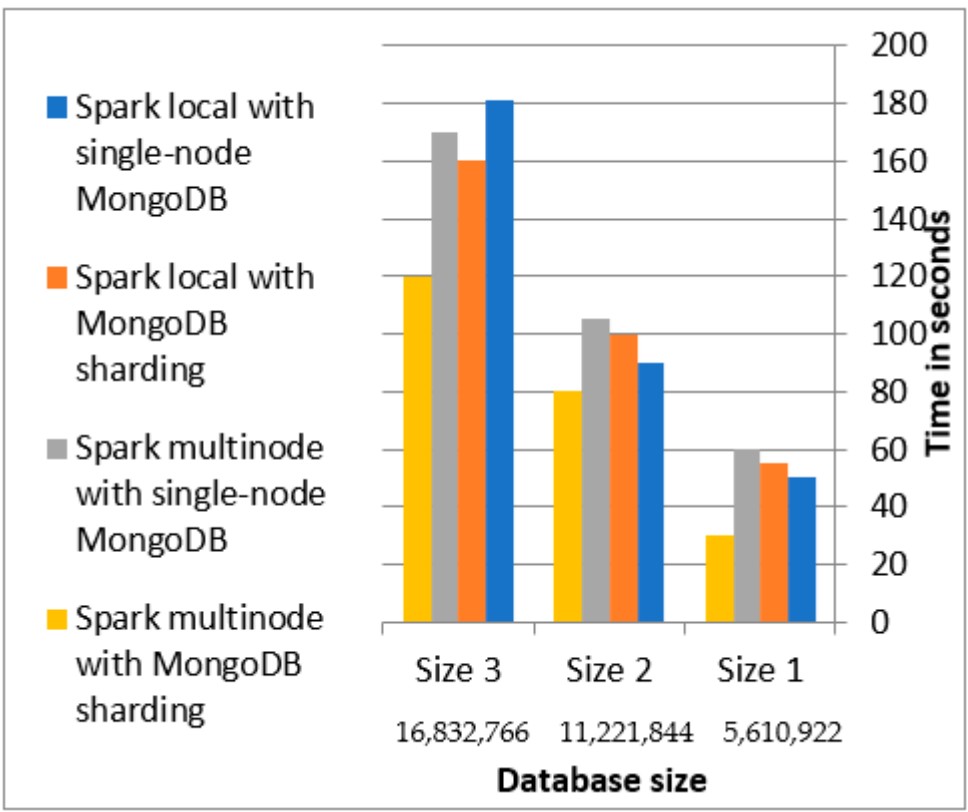

**Figure 9.** Elapsed time for executing query on MongoDB using the proposed layer.

4.2.3. Column Database Evaluation

In this subsection, the migration of data from the RDB to Cassandra is evaluated using the proposed layer. We configured a multi-node Cassandra cluster with three nodes, and one node was selected as the seed of the cluster. To achieve data locality, Cassandra nodes were installed on the same machines where the Spark workers worked. A Spark standalone master was installed on the machine, and three workers were installed on three different machines. We compared the elapsed time for migrating data from the RDB using Spark local with single-node Cassandra, Spark local with multi-node Cassandra, Spark multi-node with single-node Cassandra, and Spark multi-node with multi-node Cassandra. The parallelizability of the writes was increased by applying the following:

1.  Sorting the data before persisting it in the Cassandra database, which reduced the time of migration by 70% than without sorting the data.
2.  Spark partitions were also tuned using repartition and coalescence methods.

Figure 10 shows the elapsed time for migrating the RDB to Cassandra. From the experiments in Figure 10, the Spark multi-node with Cassandra multi-node outperformed the other experiments in the migration process. In the database with size 3, Spark multi-node with Cassandra multi-node outperformed Spark multi-node with Cassandra single node, Spark local with Cassandra multi-node, and Spark local with Cassandra single node by 45%, 41%, and 52%, respectively. The reason for the excellence of Spark Multi-node with Cassandra compared with the others is that Cassandra is a peer-to-peer distributed system, and each node can accept write operations and the installation of Spark workers and Cassandra nodes on the same machine.

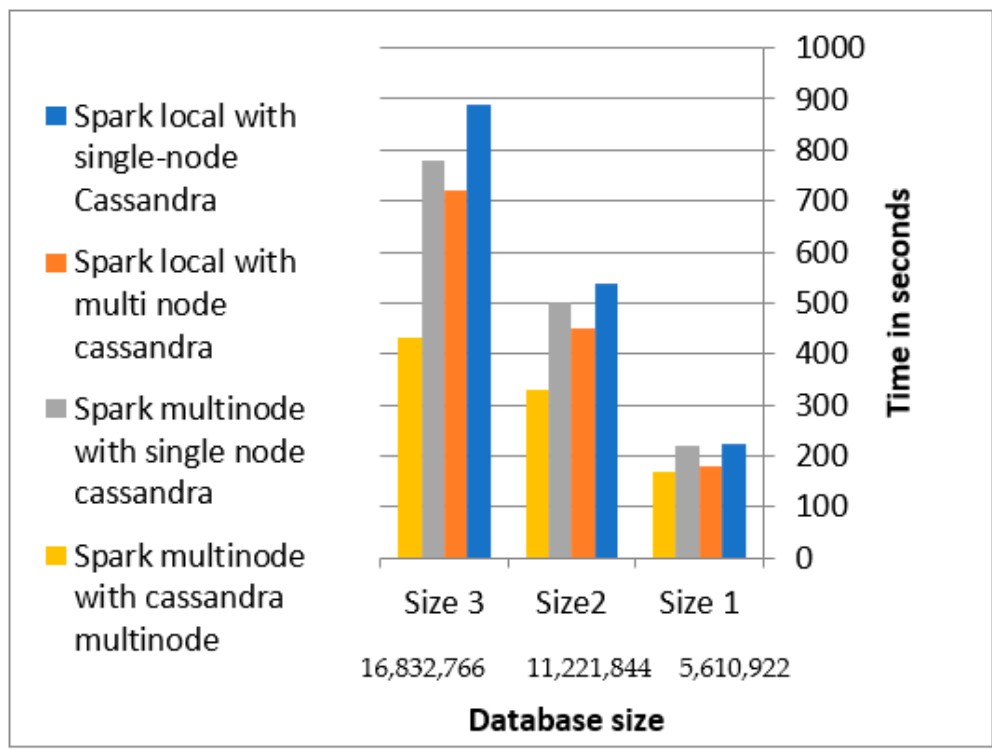

**Figure 10.** Elapsed time for migrating data to Cassandra using the proposed layer.

To evaluate the performance of the query Cassandra, we used the same query that was used to compare our proposed layer with Unity in different scenarios. As depicted in Figure 11, executing SQL using the Spark cluster with the Cassandra multi-node cluster results in the fastest processing time. Spark multi-node with Cassandra multi-node outperformed Spark multi-node with Cassandra single node, Spark local with Cassandra multi-node, and Spark local with Cassandra single node by 23%, 16%, and 31%, respectively. The reason for the excellence of the Spark cluster with Cassandra multi-node is that the Spark nodes open multiple concurrent connections to Cassandra instances.

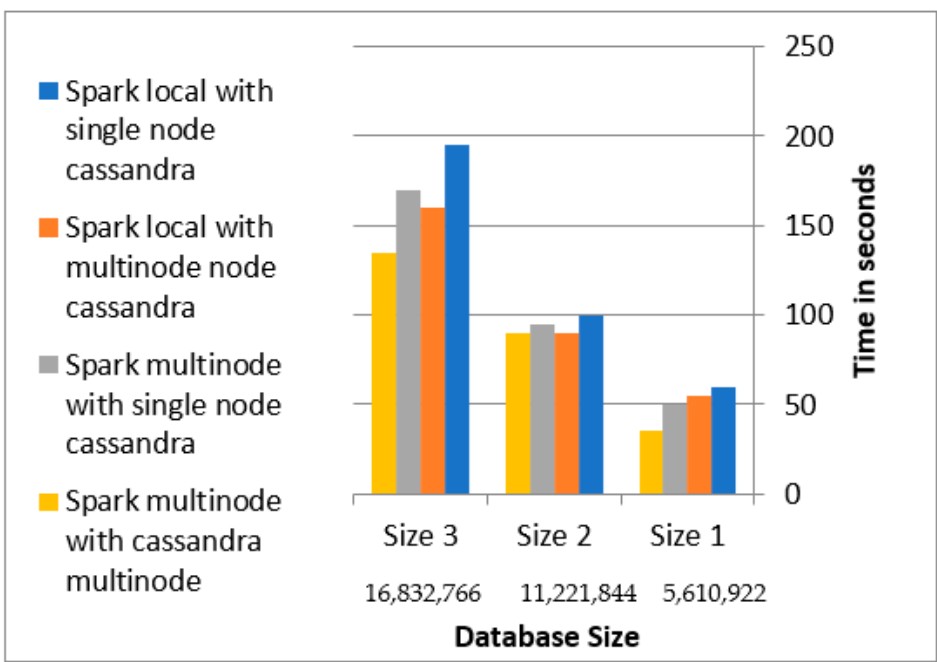

**Figure 11.** Elapsed time for executing query on Cassandra using the proposed layer.

4.2.4. Key–Value Database Evaluation

To evaluate the performance of Redis, we installed a Redis cluster with three nodes. The Spark standalone cluster was installed on four nodes; the Spark master node in a dedicated machine and the worker nodes run on the machines where the Redis cluster runs. The Redis data storage is an in-memory database. It performs well when the data fits in memory. This was limited by the overall amount of RAM. Figure 12 shows the elapsed time required for migration. In size 3, Spark multi-node with Redis cluster beats Spark multi-node with single-node Redis cluster, Spark local with multi-node Redis cluster, and Spark local with single-node Redis by 31%, 19%, and 25%, respectively.

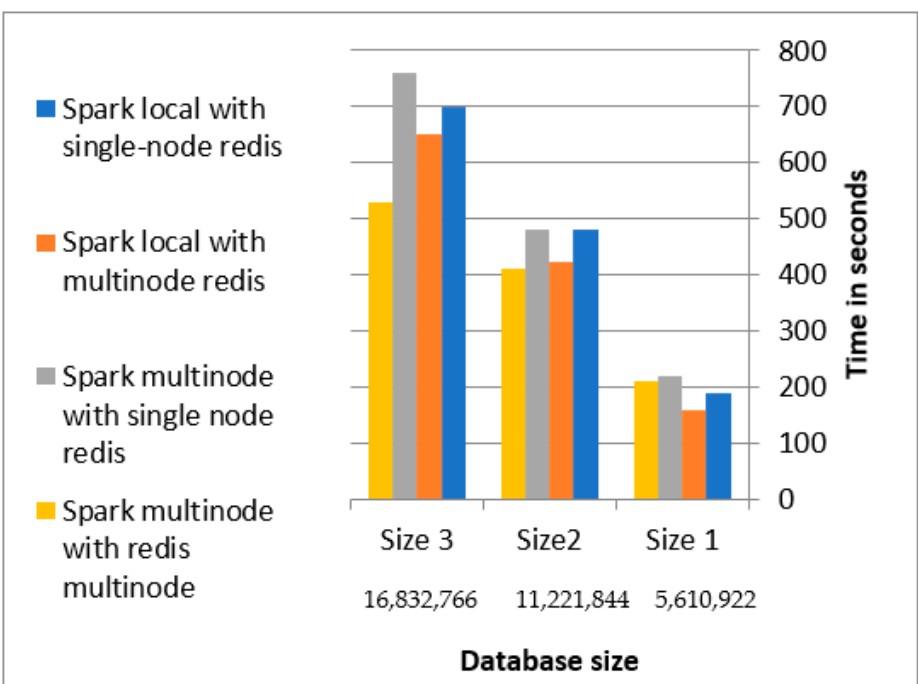

**Figure 12.** Elapsed time for migrating data to Redis using the proposed layer.

The performance when running an SQL statement query on Redis is shown in Figure 13. The performance of the Spark cluster with the Redis cluster was better than those of the other scenarios because the Spark Redis cluster opened multiple concurrent connection pools to the Redis cluster. Spark multi-node cluster with multi-node Redis cluster outperforms the Spark multi-node cluster with single Redis cluster, Spark local with multi-node Redis cluster, and Spark local with single-node Redis by 12%, 20%, and 34%, respectively, when compared with database size 3.

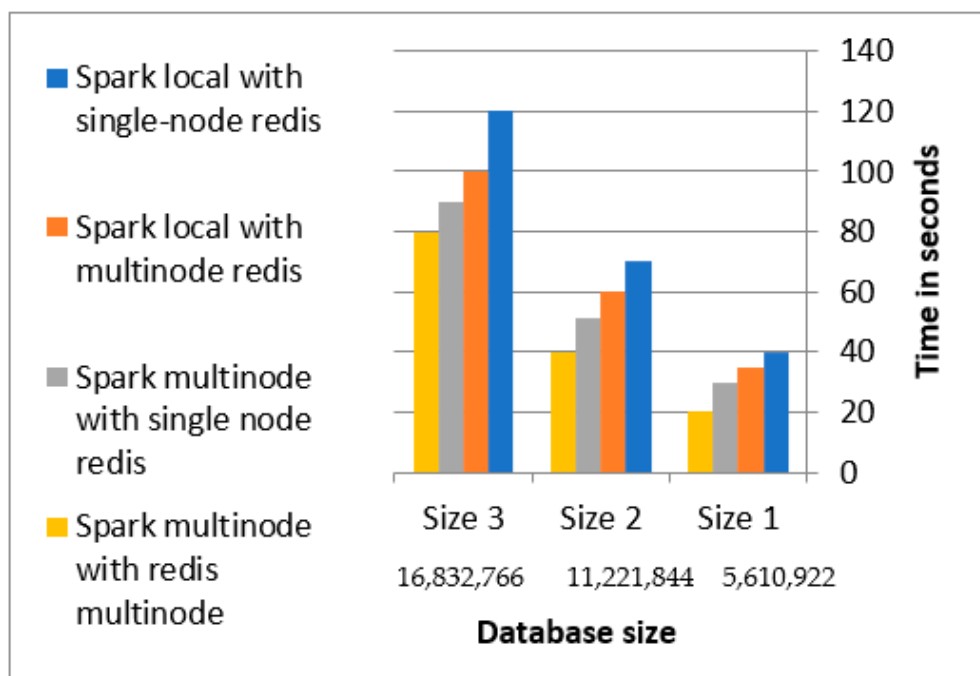

**Figure 13.** Elapsed time for executing query on Redis using the proposed layer.

4.2.5. Evaluation for Writing on Stack Exchange Dataset

The performance of migrating data from MySQL to Cassandra and MongoDBs was evaluated. As shown in Figure 14, data migration using the proposed layer on Cassandra is faster than migration to the MongoDB because the Cassandra cluster is peer-to-peer in which each instance in the cluster can receive write operations. The performance of writing data to Cassandra was faster than writing to MongoDB by 8%.

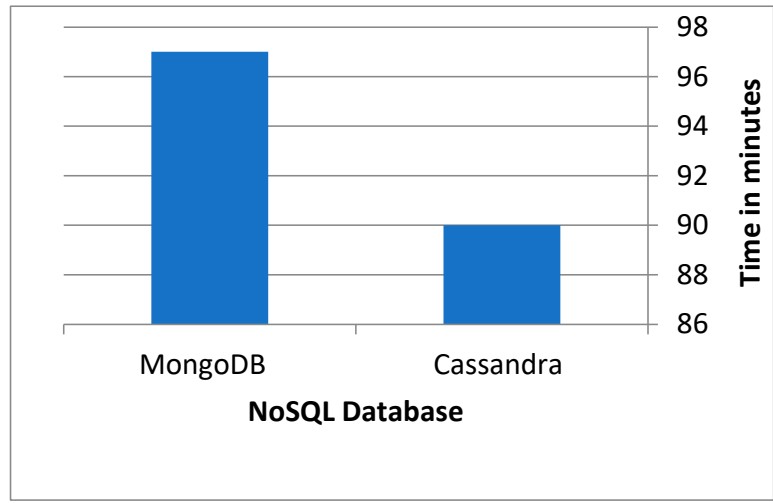

**Figure 14.** Migrating data using the proposed layer to Cassandra and MongoDB on the Stack Exchange dataset.

## 5. Conclusions

The aim of this study was to propose a Spark-based layer for converting relational databases to NoSQL models. The Spark-based layer is divided into two parts: converting relational databases to NoSQL and executing SQL on document, column, and key–value databases. The main objective when converting relational databases to NoSQL is to speed up the migration process using Spark. In addition, transformation algorithms are provided for each type of document, column, and key–value database. Each algorithm encompasses a set of rules to convert relational databases to NoSQL databases. Moreover, the proposed Spark-based layer uses the power of Spark SQL to run SQL queries that contain joins, subqueries, and aggregate functions in document, column, and key–value databases. It also translates INSERT, UPDATE, and DELETE statements to their equivalents in NoSQL databases.

The processing time of the proposed layer was evaluated against the Unity layer in a single node. The experiment was executed for SQL queries that join four tables with three different sizes of IMDB, namely, IMDB size 1 is 5,610,922 tuples, IMDB size 2 is 11,221,844 tuples, and IMDB size 3 is 16,832,766 tuples. The results of the experiment on IMDB size 3 showed that the proposed Spark-based layer outperformed Unity in terms of the query execution time by a factor of three. In addition, the proposed layer was applied to multi-node clusters using different scenarios for each NoSQL model, and the results showed that the integration between the Spark cluster and NoSQL databases on multi-node clusters provided better performance in reading and writing while increasing the dataset size than using a single node.

As for future work, the proposed Spark-based layer will be extended to be an intelligent model, so that it would be able to select a suitable NoSQL database for a given relational database.

**Author Contributions:** Formal analysis, M.A.A.-F.; Investigation, S.A.; Methodology, W.M.; Software, W.M. and S.A.; Supervision, M.A.A.-F.; Validation, S.A.; Writing—original draft, S.A. All authors have read and agreed to the published version of the manuscript.

**Funding:** This research received no external funding.

**Institutional Review Board Statement:** Not applicable.

**Informed Consent Statement:** Not applicable.

**Data Availability Statement:** Not applicable.

**Conflicts of Interest:** The authors declare no conflict of interest.

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
