# Peer review of "A Comprehensive Spark-Based Layer for Converting Relational Databases to NoSQL"

_2504-2289, doi:10.3390/bdcc6030071_

Round 1
Reviewer 1 Report
The paper addresses an area of ​​interest in the domain of databases. The paper proposes a Spark-based layer for converting relational databases to NoSQL models. The Spark-based layer is divided into two parts: converting relational databases to NoSQL and executing SQL on document, column, and key-value databases The main aim of converting relational databases to NoSQL is to speed up the migration process. In addition, the transformation algorithms are provided for each type of document, column, and key-value database. Each algorithm encompasses a set of rules to convert relational databases to NoSQL databases.
My comments/recommendations regarding the article are:
- In a relational database, many-to-many relationships are resolved through the normalization process, which often consists of creating a new intersection entity. So these relationships should be resolved in terms of a relational database. Algorithm 1 proposes a solution of these by reference or incorporation, which should be solved when implementing the relational database.
- It does not describe what type of Select operation is being tested, it is a simple select operation or a join on two or more tables because the efficiency of the operation depends on the type of operation performed.
- It also does not describe how to transform SELECT operation into NoSQL databases for example (Mongodb, Cassandra, Redis)
- A qualitative comparison between the proposed layer and the Unity model [17] is shown in Table 6. A description more detail of the Unity model is recommended.
- For more obvious results, it is recommended to indicate in Figures 7-10 the size for the size1, size2 and size3 labels.
- Fig.14, shows the migration of data using the proposed layer on Cassandra and the migration to Mongo DB but does not show the case of Redis. A comparison with Redis would be indicated in terms of migration.
- Some figures are not clearly visible, for example, Figure 1, Figure 4, etc.
- The paper is not formatted according to the requirements of the journal
Reviewer 2 Report
This manuscript is well-written and easy to follow. The comprehensive layer, created with Spark, for SQL to NoSQL transformation is beneficial for the NoSQL community.
Some minor suggestions for this manuscript:
1) The relational database to NoSQL layer proposed in this manuscript is based on Spark but it's unclear to me why the authors choose Spark over other candidates like Presto. For example, Presto supports SQL queries on NoSQL databases and contains MongoDB, Redis, and Cassandra connectors. Are there any reasons that the proposed layer has to be built with Spark?
2) In Section 3.1.2, the authors claimed that "Frequently accessed data should be stored within the same column family as much as possible." This statement is unclear to me. What if a) frequently accessed data is very large? b) frequently accessed data is queried by a wide range of queries (may even query datasets across columns)? Will these scenarios have an impact on this statement?
3) Some minor typos:
a) Mongo DB => MongoDB
b) stand-alone => standalone (Section 4.2.4)
c) Sql => SQL, Nosql => NoSQL (Section 3.2 Part II)
Author Response
"Please see the attachment

Round 2
Reviewer 1 Report
The authors provided clarifications and completed the paper taking into account the comments/observations made, which improved its content.